# Field evaluation of rapid diagnostic tests to determine dengue serostatus in Timor-Leste

**Paul Arkell**[1]*, **Maria Tanesi**[1], **Nelia Gomes**[1], **Josefina C. Joao**[2], **Tessa Oakley**[1], **Frederico Bosco**[2], **Jennifer Yan**[1], **Nicholas S. S. Fancourt**[1], **Joshua R. Francis**[1]

**1** Menzies School of Health Research, Charles Darwin University, Dili, Timor-Leste, **2** Ministry of Health, Dili, Timor-Leste

* paul.arkell@menzies.edu.au

## Abstract

The live attenuated tetravalent CYD-TDV vaccine (Dengvaxia) is effective but has scarcely been used due to safety concerns among seronegative recipients. Rapid diagnostic tests (RDTs) which can accurately determine individual dengue serostatus are needed for use in pre-vaccination screening. This study aimed to determine the performance of existing RDTs (which have been designed to detect levels of immunoglobulin G, IgG, associated with acute post-primary dengue) when repurposed for detection of previous dengue infection (where concentrations of IgG are typically lower). A convenience sample of four-hundred-and-six participants including 217 children were recruited during a community serosurvey. Whole blood was collected by phlebotomy and tested using Bioline Dengue IgG/IgM (Abbott) and Standard Q Dengue IgM/IgG (SD Biosensor) RDTs in the field. Serum samples from the same individuals were also tested at National Health Laboratory. The Panbio indirect IgG ELISA was used as a reference test. Reference testing determined that 370 (91.1%) participants were dengue IgG seropositive. Both assays were highly specific (100.0%) but had low sensitivity (Bioline = 21.1% and Standard Q = 4.6%) when used in the field. Sensitivity was improved when RDTs were used under laboratory conditions, and when assays were allowed to run beyond manufacturer recommendations and read at a delayed time-point, but specificity was reduced. Efforts to develop RDTs with high sensitivity and specificity for prior dengue infection which can be operationalised for pre-vaccination screening are ongoing. Performance of forthcoming candidate assays should be tested under field conditions with blood samples, as well as in the laboratory.

## Author summary

Dengvaxia is effective but has scarcely been used due to safety concerns among seronegative recipients. Rapid diagnostic tests (RDTs) which can accurately determine individual dengue serostatus are therefore needed for pre-vaccination screening. This study evaluated two commercially available RDTs during a community seroprevalence survey in Timor-Leste. Field conditions were intended to replicate a pre-vaccination screening setting. Both assays were specific but had low sensitivity, which was lower than previous

**Data Availability Statement:** All relevant data are within the manuscript and its Supporting Information files.

**Funding:** This work was supported by the Department for Foreign Affairs and Trade, Australian Government [Complex Grant Agreement Number 75889, https://www.dfat.gov.au/]. Funding was received by JRF. The funders did not play any role in study design, data collection and analysis, decision to publish, or preparation of the manuscript.

**Competing interests:** The authors have declared that no competing interests exist.

studies conducted in laboratories using stored serum samples. Future candidate assays should be evaluated under field conditions before they can safely be used in pre-vaccination screening.

## Introduction

Dengue is a mosquito-borne disease caused by dengue viruses 1–4 which are endemic in more than 125 countries throughout tropical and sub-tropical regions [1,2]. Due to international travel, urbanisation and climate change, global incidence has increased rapidly during the last 50 years [3]. Half the world's population are at risk of dengue, with 100–400 million infections occurring annually [1,4]. Timor-Leste is an island nation in Southeast Asia where dengue causes seasonal outbreaks across all 13 municipalities and is a major cause of morbidity and mortality among children [5].

The live attenuated tetravalent CYD-TDV vaccine (Dengvaxia) is licensed in several dengue-endemic countries for individuals 9–45 years of age. In 2017, long-term safety data showed a 1.75-fold increased risk of hospitalisation due to dengue and severe dengue among seronegative participants who received vaccination (compared with those who were seronegative but unvaccinated) [6]. The World Health Organization (WHO) therefore recommends vaccination after individuals are screened for prior infection in most settings [7]. Rapid diagnostic tests (RDTs) which can accurately determine individual dengue serostatus are needed [8,9].

Existing RDTs which detect IgG have been designed for the diagnosis of dengue in patients with acute febrile illness and/or for differentiation of primary vs. post-primary dengue. Performance is therefore likely to have been optimised based on high concentrations of IgG associated with acute infection. The potential for these devices to be repurposed for detecting previous dengue infection in healthy individuals (where typically IgG concentrations have waned over time and are lower) has been explored in previous studies. Devices have generally been shown to have high specificity but lower sensitivity when compared to traditional serological techniques, with performance varying between assays. So far, studies of commercially available RDTs have occurred in laboratories and have exclusively analysed serum samples (and not whole blood). Table 1 summarises previously published studies which have assessed the performance of various RDTs in determining dengue serostatus in healthy individuals [10–17].

The primary aim of this study was to determine the sensitivity and specificity of two RDTs when repurposed to determine dengue seropositivity in healthy individuals. Secondary aims were to compare performance of devices when used under field conditions (i.e. whole blood tested by research nurses in the community, in the presence of participants) to performance when used under laboratory conditions (i.e. serum samples tested by laboratory scientists in a diagnostic laboratory setting), explore whether any deficiencies are likely to be in-part due to assay limits-of-detection, and explore whether performance could be enhanced by allowing assays to run beyond manufacturer recommendations and read at a delayed time-point.

## Methods

### Ethics statement

This study received ethical approval from the Research Ethics and Technical Committee of the Instituto Nacional da Saude, Timor-Leste (Reference: 875 MS-INS/DGE/IX/2021) and the Human Research Ethics Committee of the Northern Territory Department of Health and Menzies School of Health Research, Australia (Reference: 2021–4064)

**Table 1. Summary of previous studies assessing performance of rapid diagnostic tests in determining dengue serostatus.**

| Publication | Participants | Reference testing | Overall seropositivity among participants by reference test | Sample type used for index testing | Setting of index testing | Index test(s) | Use of lateral flow reader | Performance of index test | |
|---|---|---|---|---|---|---|---|---|---|
| | | | | | | | | Sens. | Spec. |
| Bonaparte M et al. (Oct 2019) [10] | 804 individuals who were participants in multiple clinical trials of CYD-TDV or employees of Sanofi Pasteur in the USA. | Serostatus for each sample defined using a 'combination of epidemiological, clinical (when available) and assay result information'. Sensitivity (positive) and specificity (negative) panels created. | 33.6% | Serum | Laboratory: Global ClinicalImmunology Laboratory, Sanofi Pasteur, Swiftwater, USA. | RDT Dengue IgA/IgG (Bio-Rad, Hercules, USA) (IgG component read) | No | 69.6% | 99.4% |
| | | | | | | OnSite Dengue IgG/IgM (CTK Biotech, San Diego, USA) (IgG component read) | No | 67.0% | 98.9% |
| | | | | | | SD Bioline Dengue IgG/IgM (Abbott, Chicago, USA) (IgG component read) | No | 53.7% | 99.6% |
| | | | | | | Dengue IgG/IgM Rapid Test (GenBody/Bahiafarma, Cheonan, Korea) (IgG component read) | No | 39.6% | 99.1% |
| Bonaparte M et al. (Mar 2020) [11] | 709 individuals from both dengue endemic and non-endemic regions who were participants in CYD-TDV (Dengvaxia) trials. | Seronegative status defined as $PRNT_{50}$-negative (non-endemic) and according to algorithm including data from: $PRNT_{50}$, $PRNT_{90}$, anti-dengue NS1 IgG ELISA (endemic). Seropositive status defined by Dengvaxia phase 3 trial. Sensitivity (positive) and specificity (negative) panels created. | 53.5% | Serum | Laboratory: A 'Puerto Rican laboratory certified under the Clinical Laboratory Improvement Amendments statute for diagnosis of acute dengue infection'. | Tell Me Fast Dengue IgG/IgM Combo Test Device (Biocan Diagnostics, Vancouver, Canada) (IgG component read) | No | 61.0% | 99.1% |
| DiazGrandos C et al. (Nov 2020) [12] | 3962 children previously included in phase 3 CYD-TDV studies including those aged 2–14 years in Asia-Pacific region and those aged 9–16 years in Latin American region. | Assigned seropositive/negative according to reference algorithm including data from: 1. $PRNT_{50}$ 2. $PRNT_{90}$ 3. Anti-dengue NS1 IgG ELISA | 69.0% | Serum | Laboratory: Central Virology Laboratory (Chaim Sheba Medical Center, Ramat Gan, Israel | SD Bioline Dengue IgG/IgM (Abbott, Chicago, USA) (IgG component read) | No | 71.1% | 96.0% |
| | | | | | Laboratory: Sanofi Pasteur's Global Clinical Immunology Laboratory, Swiftwater, USA | Tell Me Fast Dengue IgG/IgM Combo Test Device (Biocan Diagnostics, Vancouver, Canada) (IgG component read) | No | 52.5% | 99.0% |
| | | | | | | OnSite Dengue IgG/IgM Combo Rapid Test CE (CTK Biotech, San Diego, USA) (IgG component read) | No | 47.6% | 99.5% |

*(Continued)*

**Table 1.** (Continued)

| Publication | Participants | Reference testing | Overall seropositivity among participants by reference test | Sample type used for index testing | Setting of index testing | Index test(s) | Use of lateral flow reader | Performance of index test | |
|---|---|---|---|---|---|---|---|---|---|
| | | | | | | | | Sens. | Spec. |
| Chong ZL et al. (Feb 2021) [13] | 484 individuals in communities covered by District Health Office dengue vector control activities. Mean (SD) age in years = 32.0 (18.4) | Two or more positive tests required from: 1. Panbio Dengue IgG Indirect ELISA (Abbott, Chicago, USA) 2. In-house HI assay 3. In-house FRNT | 79.1% | Capillary blood | Field: Two communities in Petaling District, Selangor State, Malaysia | ViroTrack Dengue Serostate (BluSense Diagnostics, Copenhagen, Denmark) (Detects IgG) | N/A | 91.1% | 91.1% |
| Daag JV et al. (Jun 2021) [14] | 1000 healthy children aged 9–14 years in a cohort study in Bogo and Balamban in the Philippines | mFRNT | 91.4% | Serum | Laboratory: University of the Philippines Manila National Institutes of Health | Ichroma II (Boditech Med Incorporated, Gang-won-do, Republic of Korea) (Detects IgG) | N/A | 91.8% | 90.7% |
| Echegaray F et al. (July 2021) [15] | 24 individuals aged 18–93 years who had confirmed dengue or zika virus infection between 1–30 years prior | FRNT | 66.7% | Blood | Laboratory: Unclear exact locations in USA and Ecuador | SD BIOLINE Dengue IgG/IgM RDT (Bitrodiagnostico, Quito, Ecuador) (IgG component read) | No | 0.0% | 100.0% |
| | 69 participants in the EcoDess cohort in Borbón, Ecuador (aged 2–60 years). | Dengue PanBio IgG ELISA, using cutoff values established by DENV1-4 PRNT titres: - DENV-naïve (<0.2 units) - DENV-primary (0.2–0.9) - DENV-secondary (>0.9) | 63.8% | Serum | | | | 40.9% | 100.0% |
| | 52 children aged 2–17 years from cohort studies in Managua, Nicaragua, who had confirmed dengue infections 6–12 months prior. | Initial infection confirmed by RT-PCR, virus isolation, or a serological algorithm. Serostatus confirmed by testing paired annual samples using the 'dengue inhibition ELISA' and/or 'reporter-virus based neutralization assay'. | 63.5% | Serum/ plasma | Laboratory: Unclear exact locations in Nicaragua and/or Ecuador | Excivion Dengue RDT (Excivion Ltd, Cambridge, UK) (Detects IgG) | No | 78.8% | 78.9% |
| | 115 children aged 8 in the ECUAVIDA birth cohort in Ecuador | Abcam Human Anti-Dengue virus IgG ELISA Kit. | 65.2% | Serum | | | | 63.5%* | >= 90%* |

(*Continued*)

**Table 1.** (Continued)

| Publication | Participants | Reference testing | Overall seropositivity among participants by reference test | Sample type used for index testing | Setting of index testing | Index test(s) | Use of lateral flow reader | Performance of index test | |
|---|---|---|---|---|---|---|---|---|---|
| | | | | | | | | Sens. | Spec. |
| Limothai et al. (Sep 2021) [16] | Healthy children aged 9–22 years in Ratchaburi province, Thailand. | PRNT | 81.7% | Serum | Laboratory: Central Laboratory, Chulalongkorn hospital, Thailand | SD Bioline Dengue Duo NS1 Ag & IgG/IgM (SD Bioline, Korea) (IgG component read) | No | 35.1% | 100.0% |
| Savarino et al. (May 2022) [17] | 3833 children previously included in phase 3 CYD-TDV studies including those aged 2–14 years in Asia-Pacific region and those aged 9–16 years in Latin American region. | Assigned seropositive/negative according to reference algorithm including data from: 1. PRNT$_{50}$ 2. PRNT$_{90}$ 3. Anti-dengue NS1 IgG ELISA | 68.7% | Serum | Laboratory: Global Clinical Immunology Laboratory (Sanofi Pasteur, Swiftwater, PA, USA) | OnSite Dengue IgG RDT (Onsite IgG RDT; CTK Biotech, Poway, CA, USA) (Detects IgG) | No | 91.1% | 92.8% |
| Arkell et al. (Oct 2022) (This study) | 189 healthy adults and 217 healthy children in Dili Municipality, Timor-Leste who were part of a population-representative serosurvey | Dengue PanBio Indirect IgG ELISA using kit cut-offs | 91.1% | Whole venous blood | Field: Tests operated and interpreted by research nurses in the community, in the presence of participants | SD Bioline Dengue IgG/IgM (Abbott, Chicago, USA) (IgG component read) | No | 21.1% | 100.0% |
| | | | | | | Standard Q Dengue IgM/IgG (SD Biosensor, Republic of Korea) (IgG component read) | No | 4.6% | 100.0% |

**Abbreviations:** Sens. = sensitivity, spec. = specificity, PRNT$_{50}$ = plaque reduction neutralisation test with 50% cut-off, PRNT$_{90}$ = plaque reduction neutralisation test with 90% cut-off, ELISA = enzyme-linked immunosorbent assay, NS1 –non-structural protein 1, HI = hemagglutination inhibition, FRNT = focus reduction neutralisation test, mFRNT = micro focus reduction neutralization test

## Participants

A population-representative sample of individuals aged one year or older living in Dili was identified as part of a three-stage cluster sampling design for a national seroprevalence study of vaccine-preventable diseases (including dengue) in Timor-Leste. Potential participants were visited at their homes. The first 200 adults and first 200 children (aged < 16 years) who were recruited within Dili municipality were also included in this sub-study, evaluating the performance of dengue RDTs. This sample size was considered adequate to estimate 70% sensitivity and 95% specificity of RDTs with 5% precision (alpha = 0.05) [18]. Since there has been no previous population serological surveillance of dengue in Dili, seroprevalence was assumed to be 80% (i.e. similar to other urban Southeast Asian settings with high burden of dengue) [19].

## Sample collection

Research nurses collected blood from participants by phlebotomy using either a standard hypodermic or a winged butterfly needle with a syringe attached. A small volume of whole blood was applied to the index test RDTs (methods below), while most of the sample was

injected into a gel serum (SST) tube, which underwent centrifugation in the field (1,500 RCF, 10 minutes). Separated serum samples were transported to the National Health Laboratory (NHL) in Dili within a portable refrigerator and then stored at 4 degrees Celsius until analysis.

### Reference test

Serum samples were analysed using the Panbio indirect IgG ELISA (product code 01PE30). This assay was chosen because it has been has been designed for detection of previous dengue (as opposed to diagnosis of acute dengue infection), it has been used in serosurveillance studies previously [20,21], it has acceptable performance when compared to virus neutralisation testing (considered most accurate in determining dengue serostatus and correlates best with outcomes after Dengvaxia) [12,22], and because laboratory equipment and expertise for its use were available at the NHL. Testing was performed in accordance with manufacturers' instructions: A cut-off value was calculated for each run by multiplying the calibration factor (batch-specific) by the mean optical densities (ODs) of calibrator material (run in triplicate). An index value (IV) was then calculated by dividing the sample OD by the cut-off value. Samples with IV < 0.9 were assigned 'dengue IgG negative'. Samples with index value > 1.1 were assigned dengue IgG positive. Samples with IV 0.9–1.1 were repeat-tested with the second result being used. If IV remained 0.9–1.1 on repeat testing then the sample was assigned 'dengue IgG positive'. The appropriateness of these commercial serological cut-offs was assessed visually by plotting a histogram showing ELISA IVs from all serum samples in the study, which was assessed visually (see S1 Fig). The cut-off values appeared to adequately distinguish two populations of antibody responses and were therefore adopted into the study.

### Testing with RDTs

Participants were tested for the presence of dengue antibodies using two different commercially available dengue RDTs (Bioline Dengue IgG/IgM, Abbott and Standard Q Dengue IgM/IgG, SD Biosensor). These were chosen based their positive performance in previous studies (Bioline sensitivity 53.7–71.1%, specificity 96.0–99.6%) [10,12], and their availability and low cost in Timor-Leste (Standard Q). Initially, RDTs were used under field conditions (i.e. whole blood was tested by research nurses in the community, in the presence of participants). This was intended to mimic a potential pre-vaccination screening setting. Subsequently, RDTs were used under laboratory conditions (i.e. serum samples from the same individuals were tested by laboratory scientists at NHL, which is a diagnostic laboratory).

In both test settings RDTs were operated in accordance with manufacturers' instructions. Each device was labelled with the participant ID number and placed on a flat surface. In the field they were inoculated with two drops of whole blood from the syringe. In the laboratory, they were inoculated with two drops of serum from the SST tube. A timer was set for 15 minutes, after which the IgG and IgM lines were interpreted visually by a single researcher and reported in real-time. For the laboratory (serum) experiment, the assays were also read at 60 minutes, to assess whether this would affect test performance. Research nurses and laboratory scientists attended classroom teaching sessions on the safe and appropriate use the RDTs which included practicing operating each device and being observed, which was delivered by PA.

### Participant management and ethical considerations

Written consent was obtained from the participants and also from the parent/guardian (in the case of children). Participants were informed of their RDT results in the field. If dengue IgG was detected by either assay (but IgM was not detected), participants were informed that they

had likely been infected with dengue at some time in their lives, but likely not recently. If IgM (+/- IgG) was detected by either assay, participants were informed that there was evidence of current or recent dengue infection. These individuals were advised to seek medical attention if they experienced fever or other acute illness during the subsequent two weeks. These interpretations were agreed by members of the research team who have backgrounds in medical microbiology, adult, and paediatric infectious disease (PA, JY and JF). They are consistent with interpretations suggested by the RDT instructions for use, with the exception of isolated IgG seropositivity which was considered significantly more likely to represent previous infection than acute dengue in this group of asymptomatic participants.

## Statistical analysis

The sensitivity and specificity of each RDT for determining IgG seropositivity under each set of operating conditions was assessed against the reference test. To explore whether RDT insensitivity was likely to be in-part due to an assay limit-of-detection (LOD), ELISA IVs from reference testing were analysed as a proxy for antibody concentrations. RDT true positive samples were compared to RDT false negative samples and unpaired Students T-tests were used to determine whether there was a significant difference. Results were considered significant where $p < 0.05$.

## Results

Four-hundred-and-six individuals were included in this study. This included 217 children whose median (interquartile range, IQR) age in years was 10 (6–13) and 189 adults whose median (IQR) age in years was 33 (24–42). Reference testing determined that 186/217 (85.7%) children and 184/189 (97.4%) adults were dengue IgG seropositive (overall seropositivity = 91.1%).

## Performance of RDTs when testing blood samples under field conditions

The sensitivity and specificity of the Bioline assay when testing blood under field conditions (interpreted after 15 minutes, as per manufacturers' instructions) were 21.1% and 100.0%, respectively. The sensitivity and specificity of the Standard Q under similar conditions were 4.6% and 100.0%, respectively.

## Performance of RDTs when testing serum under laboratory conditions

The sensitivity and specificity of the Bioline assay when testing serum under laboratory conditions (interpreted after 15 minutes, as per manufacturers' instructions) were 42.7% and 97.2%, respectively. When tests were allowed to run for 60 minutes (i.e. beyond manufacturer recommendations) sensitivity was improved to 80.3% but specificity reduced to 94.4%.

The sensitivity and specificity of the Standard Q assay when testing serum under laboratory conditions (interpreted after 15 minutes, as per manufacturers' instructions) were 10.5% and 97.2%, respectively. When tests were allowed to run for 60 minutes (i.e. beyond manufacturer recommendations) sensitivity was improved to 38.4% and specificity remained 97.2%. Table 2 summarises the performance of each RDT under different test conditions.

## Analysis of ELISA IVs as a proxy for antibody concentration

Among participants whose Bioline RDT was true positive when used under field conditions the mean (standard deviation, SD) IV of the reference ELISA test was 3.808 (0.535), compared to 3.300 (0.779) in participants whose Bioline RDT was false negative ($p < 0.001$). Among

**Table 2. Performance of rapid diagnostic tests (RDTs) when operated under different test conditions.**

| | | | Panbio indirect IgG ELISA (reference test) | | Sensitivity (95% CI) | Specificity (95% CI) | Positive predictive value (95%CI) | Negative predictive value (95%CI) |
|---|---|---|---|---|---|---|---|---|
| | | | Positive | Negative | | | | |
| Bioline Dengue IgG/IgM (Abbott) | Field 15 | Positive | 78 | 0 | 21.1% (17.2–25.52) | 100.0% (90.4–100.0) | 100.0% 95.3–100.0) | 11.0% (8.0–14.8) |
| | | Negative | 292 | 36 | | | | |
| | Lab 15 | Positive | 158 | 1 | 42.7% (37.8–47.8) | 97.2% (85.8–99.5) | 99.4% (96.5–99.9) | 14.2% (10.4–19.1) |
| | | Negative | 212 | 35 | | | | |
| | Lab 60 | Positive | 297 | 2 | 80.3% (75.9–84.0) | 94.4% (81.9–98.5) | 99.3% (97.6–99.8) | 31.8% (23.7–41.1) |
| | | Negative | 73 | 34 | | | | |
| Standard Q Dengue IgM/IgG (SD Biosensor) | Field 15 | Positive | 17 | 0 | 4.6% (2.9–7.2) | 100.0% (90.4–100.0) | 100.0% (81.6–100.0) | 9.3% (6.8–12.6) |
| | | Negative | 353 | 36 | | | | |
| | Lab 15 | Positive | 39 | 1 | 10.5% (7.8–14.1) | 97.2% (85.3–99.5) | 97.5% (87.12–99.7) | 9.6% (7.0–13.0) |
| | | Negative | 331 | 35 | | | | |
| | Lab 60 | Positive | 142 | 1 | 38.4% (33.6–43.4) | 97.2% (85.8–99.5) | 99.3% (96.2–99.9) | 13.3% (9.7–18.0) |
| | | Negative | 228 | 35 | | | | |

Abbreviations: Field = field conditions, Lab = laboratory conditions, 15 = interpreted after 15 minutes, 60 = interpreted after 60 minutes, ELISA = enzyme-linked immunosorbent assay

participants whose Standard Q test was true positive when used under field conditions the mean (SD) ELISA IV of the reference ELISA test was 3.740 (0.441), compared to 3.392 (0.772) in participants whose Standard Q RDT was false negative (p = 0.007). This indicates the RDTs were more likely to accurately identify positive samples which have higher dengue IgG concentrations and hence RDT insensitivity is at least partially due to an assay limit-of-detection problem. Fig 1 shows ELISA IVs in each of these groups. Table 3 summarises similar

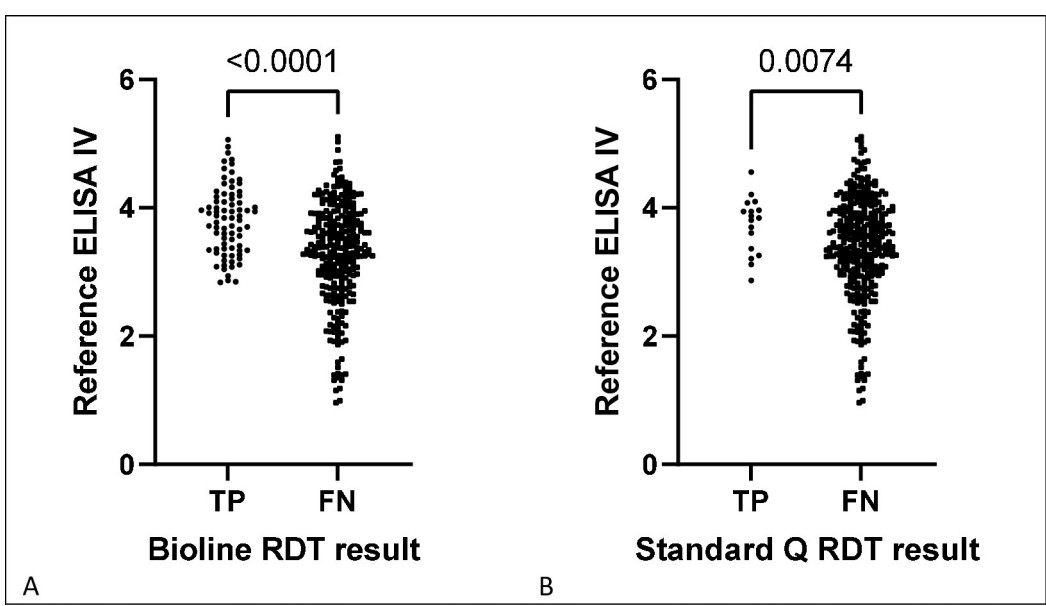

**Fig 1.** Comparison of reference enzyme-linked immunosorbent assay (ELISA) index values (as a proxy for antibody concentration) between individuals who were rapid diagnostic test (RDT) -true positive (TP) and RDT-false negative (FN) when Bioline Dengue IgG/IgM, Abbott (A) and Standard Q Dengue IgM/IgG, SD Biosensor (B) assays were used under field conditions.

**Table 3. Comparison of mean reference ELISA index values (IVs) between individuals who had true positive vs. false negative rapid diagnostic tests (RDTs) under differing test conditions.**

|  |  | True positive | False negative | p = [*] |
|---|---|---|---|---|
| Bioline Dengue IgG/IgM (Abbott) | Field | 3.808 (0.535) | 3.300 (0.779) | < 0.001 |
|  | Lab 15 | 3.780 (0.601) | 3.130 (0.753) | < 0.001 |
|  | Lab 60 | 3.521 (0.679) | 2.944 (0.903) | < 0.001 |
| Standard Q Dengue Dengue IgM/IgG (SD Biosensor) | Field | 3.740 (0.441) | 3.392 (0.772) | 0.007 |
|  | Lab 15 | 4.099 (0.517) | 3.326 (0.746) | < 0.001 |
|  | Lab 60 | 3.760 (0.629) | 3.188 (0.757) | < 0.001 |

[*] Unpaired Student's T-tests were used to assess differences in IVs between groups

Abbreviations: Field = field conditions, Lab = laboratory conditions, 15 = interpreted after 15 minutes,

60 = interpreted after 60 minutes

comparisons of ELISA IVs across different test conditions. Study data are available in the supplementary information (S1 Data).

## Discussion

This study evaluated the performance of two RDTs in determining dengue IgG seropositivity among healthy individuals (using serum ELISA as the reference test). It is the first study in which commercially available RDTs have been evaluated for this purpose using blood samples, and the first such study conducted under field conditions intended to replicate a pre-vaccination screening setting. When operated according to manufacturers' instructions (i.e. read at 15 minutes), the Bioline and Standard Q assays were highly specific (both 100.0%) but had low sensitivity (21.1% and 4.6%, respectively). These results contrast with previous studies which have tested serum samples under laboratory conditions and found sensitivity of the Bioline assay to be 53.7–71.1%.[10,12] Indeed, when RDTs were operated under laboratory conditions using serum samples from the same individuals in this study, the sensitivity of both assays appeared improved (42.7% and 10.5%, respectively). When tests were allowed to run for 60 minutes (i.e. beyond manufacturer recommendations) sensitivity was further improved (80.3% and 38.4% respectively), but this appeared to be at the expense of specificity (which reduced to 97.2% and 94.4%, respectively). Similar observations were made in a recent study which aimed to optimise both commercially available and purpose-designed RDTs by varying run-time [15].

This study also found that mean IV of the reference ELISA test was consistently significantly higher in participants whose RDTs were true positive compared with those whose RDTs were false negative, indicating that RDT insensitivity is at least partially due to a limit-of-detection problem with these two RDTs. This is consistent with assays having been developed for the diagnosis of dengue in patients with acute febrile illness and differentiation of primary vs. post-primary dengue (where concentrations of IgG are usually higher compared to those seen long after convalescence). Previous studies have found that some ELISAs which detect dengue IgG can also be insensitive, particularly in individuals who have lower titres of neutralising antibodies and/or where there is a monotypic antibody profile [22,23]. Importantly, the latter group of individuals are most likely to benefit most from vaccination but would be denied this if pre-vaccination screening was falsely negative.

Additional potential sources of low RDT sensitivity include the following: First, RDTs may be interpreted too soon after inoculation. This is supported by data from the present study

which showed a difference in sensitivity when time-to-interpretation was varied. Second, there may be difficulties in the visual interpretation of RDTs, particularly when operators are working under field conditions (where the level of lighting may be low). Fieldworkers in the present study received classroom training in the use of both RDTs and were observed to ensure proper technique, however there was no formal programme of quality assurance to ensure proficiency in visual interpretation. The use of lateral flow readers and/or mobile phone app readers [24], including those which verify run-time (or would not make a determination before the appropriate run-time has elapsed), may improve performance. Larger field studies which assess performance at fine temporal resolutions to determine the optimal run-time may be useful. Third, while the chosen reference test (Panbio indirect IgG ELISA) has been designed for detection of previous dengue (as opposed to diagnosis of acute infection), and has been used in serosurveillance studies previously [20,21], it may have mis-assigned some samples. This could have occurred because the commercial serological cut-off was used rather than any population-specific alternative derived from the observed antibody responses. While the appropriateness of cut-offs were assessed visually, the sample size for this study was not considered large enough to undertake mixture modelling [25,26]. Furthermore, virus neutralisation assays may have been more accurate in determining dengue serostatus (and could correlate better with outcomes after Dengvaxia) than ELISAs [12,22], however they were not available during this study which took place in Timor-Leste.

Although this study found RDTs to have high specificity, false positive results were observed. These could have been due to antibodies against other flaviviruses which may be circulating in Timor-Leste including zika and Japanese encephalitis viruses, which commonly cross-react in serological assays [27]. If used in pre-vaccination screening, RDTs which give false positive results would give rise to inappropriate vaccine administration. As such, it would be important to further characterise any false positive samples from candidate assays and perform more extensive specificity testing using panels of samples which are known to have potentially cross-reacting antibodies.

This study identified high dengue seroprevalence among both adults and children which is consistent with other studies from the Southeast Asian region. A previous study from Timor-Leste identified lower seroprevalence, but that study tested dried blood spots from individuals in rural areas and used a less sensitive ELISA [28]. Results from the ongoing population-representative serosurvey (from which this study took a convenience sample) will therefore be crucial in determining the most appropriate disease control strategy. If seroprevalence in the vaccine candidate age-group is very high (>70%), universal administration of Dengvaxia without pre-vaccination screening may be appropriate, because the number of severe cases prevented in those who were seropositive would likely be substantially greater than the excess number that occurred in those who were seronegative [7].

Dengue is an emerging infection which causes significant morbidity and mortality across tropical and sub-tropical regions. However, the only effective vaccine has scarcely been used because of safety concerns among seronegative recipients. Efforts to develop an assay with high sensitivity and specificity for prior dengue infection which can be operationalised for pre-vaccination screening are ongoing. Performance of forthcoming candidate assays should be tested under field conditions with blood samples (as well as in the laboratory).

## Supporting information

**S1 Data. Study data.**
(XLSX)

**S1 Fig. Histogram showing antibody responses in serum samples from all individuals with commercial cut-off values shown.**
(PDF)

## Author Contributions

**Conceptualization:** Paul Arkell, Josefina C. Joao, Tessa Oakley, Frederico Bosco, Jennifer Yan, Nicholas S. S. Fancourt, Joshua R. Francis.

**Data curation:** Paul Arkell, Tessa Oakley.

**Formal analysis:** Paul Arkell, Nelia Gomes, Tessa Oakley, Nicholas S. S. Fancourt, Joshua R. Francis.

**Funding acquisition:** Paul Arkell, Jennifer Yan, Joshua R. Francis.

**Investigation:** Paul Arkell, Maria Tanesi, Nelia Gomes, Josefina C. Joao, Jennifer Yan, Nicholas S. S. Fancourt, Joshua R. Francis.

**Methodology:** Paul Arkell, Maria Tanesi, Nelia Gomes, Josefina C. Joao, Tessa Oakley, Frederico Bosco, Jennifer Yan, Nicholas S. S. Fancourt, Joshua R. Francis.

**Project administration:** Paul Arkell, Maria Tanesi, Nelia Gomes, Josefina C. Joao.

**Resources:** Paul Arkell, Maria Tanesi, Nelia Gomes, Frederico Bosco, Jennifer Yan, Joshua R. Francis.

**Software:** Paul Arkell.

**Supervision:** Paul Arkell, Tessa Oakley, Jennifer Yan, Nicholas S. S. Fancourt, Joshua R. Francis.

**Validation:** Paul Arkell, Maria Tanesi, Nelia Gomes.

**Visualization:** Paul Arkell.

**Writing – original draft:** Paul Arkell.

**Writing – review & editing:** Maria Tanesi, Nelia Gomes, Josefina C. Joao, Tessa Oakley, Frederico Bosco, Jennifer Yan, Nicholas S. S. Fancourt, Joshua R. Francis.

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
