## [Decision Letter · Decision Letter 0]

6 Sep 2022

Dear Dr Arkell

Thank you very much for submitting your manuscript "Field evaluation of rapid diagnostic tests to determine dengue serostatus in Timor-Leste" for consideration at PLOS Neglected Tropical Diseases. As with all papers reviewed by the journal, your manuscript was reviewed by members of the editorial board and by several independent reviewers. The reviewers appreciated the attention to an important topic. Based on the reviews, we are likely to accept this manuscript for publication, providing that you modify the manuscript according to the review recommendations. 

Sincerely,

Renata Rosito Tonelli, PhD

Academic Editor

Elvina Viennet

Section Editor

Reviewer's Responses to Questions

**Key Review Criteria Required for Acceptance?**

**Methods**

-Are the objectives of the study clearly articulated with a clear testable hypothesis stated?

-Is the study design appropriate to address the stated objectives?

-Is the population clearly described and appropriate for the hypothesis being tested?

-Is the sample size sufficient to ensure adequate power to address the hypothesis being tested?

-Were correct statistical analysis used to support conclusions?

-Are there concerns about ethical or regulatory requirements being met?

Reviewer #1: Yes

Reviewer #2: This manuscript describes a study to assess the performance of two rapid diagnostic tests (RDTs) to determine dengue serostatus for pre-vaccination screening under field and laboratory conditions in Timor-Leste. The RDTs were performed using whole blood collected from 406 participants, of whom 217 were children. The results were compared to serum samples from the same individuals and tested at the National Health Laboratory using an enzyme-linked immunosorbent assay (ELISA) on serum samples as the comparator. The authors found that the RDTs had low sensitivity which can be improved by using serum and by extending the window for reading the results.

This is a well written paper on a topic that is critically important for the deployment of the only approved dengue vaccine, that can save lives. The study objectives are clearly stated, the study was well designed and the methods appropriate for the study objectives. 

There are no ethical or regulatory issues.

**Results**

-Does the analysis presented match the analysis plan?

-Are the results clearly and completely presented?

-Are the figures (Tables, Images) of sufficient quality for clarity?

Reviewer #1: Yes

Reviewer #2: The results are clearly presented in tables that are easy to understand and interpret and figures that are good quality. 

As the interpretation of RDT results are subjective, the authors should previde details on training of field staff to perform the tests and whether external quality assessment was carried out to ensure proficiency of staff in performing the tests and interpreting the test results.

**Conclusions**

-Are the conclusions supported by the data presented?

-Are the limitations of analysis clearly described?

-Do the authors discuss how these data can be helpful to advance our understanding of the topic under study?

-Is public health relevance addressed?

Reviewer #1: Yes

Reviewer #2: The conclusions are supported by the data but there are a few points that should be discussed in more detail:

1. Reference standard: As there is no reference materials for determining dengue seropositivity, the choice of the assay as the comparator clearly affects the test performance. As shown in Table 3, previous studies use different reference standards, the authors should discuss the different reference standards used in previous studies and how they may affect the study results.

2. Discussion: It is well known that IgG antibody responses to different flaviviruses are known to be cross-reactive. The authors should discuss the implications of false positive and false negative results from the use of the RDTs to determine dengue serostatus. 

3. The WHO Strategic Advisory Group of Experts (SAGE) on Immunization issued recommendations in April 2016 to introduce the dengue vaccine (Dengvaxia®) in geographical settings (national or subnational) with high endemicity only, as indicated by seroprevalence of >70% in the age group targeted for vaccination. The findings from this study population showed seropositivity rates in excess of 85%, the authors should discuss whether there is still a need for Timor Leste to conduct pre-screening for vaccine introduction.

**Editorial and Data Presentation Modifications?**

Reviewer #1: Minor Revision

Reviewer #2: See above

**Summary and General Comments**

Reviewer #1: (No Response)

Reviewer #2: The evaluation of the performance of dengue RDTs to determine dengue serostatus is not new but the data on possible reasons for low sensitivity and how to improve it are useful.

PLOS authors have the option to publish the peer review history of their article (what does this mean?). If published, this will include your full peer review and any attached files.

Reviewer #1: Yes: Dr Joseph Biggs

Reviewer #2: No

Figure Files:

Data Requirements:

Reproducibility:

References

---

## [Editor Report · Decision Letter 1]

7 Oct 2022

Dear Dr Ankel

We are pleased to inform you that your manuscript 'Field evaluation of rapid diagnostic tests to determine dengue serostatus in Timor-Leste' has been provisionally accepted for publication in PLOS Neglected Tropical Diseases.

Best regards,

Renata Rosito Tonelli, PhD

Academic Editor

Elvina Viennet

Section Editor

---

## [Editor Report · Acceptance letter]

29 Oct 2022

Dear Dr Arkell,

We are delighted to inform you that your manuscript, "Field evaluation of rapid diagnostic tests to determine dengue serostatus in Timor-Leste," has been formally accepted for publication in PLOS Neglected Tropical Diseases.

Best regards,

Shaden Kamhawi

co-Editor-in-Chief

Paul Brindley

co-Editor-in-Chief
